# Conditioned Implicit Neural Representation for Regularized Deformable Image Registration

## Abstract

Regularization is essential in deformable image registration (DIR) to ensure that the estimated Deformation Vector Field (DVF) remains smooth, physically plausible, and anatomically consistent. However, fine-tuning regularization parameters in learning-based DIR frameworks is computationally expensive, often requiring multiple training iterations. To address this, we propose cIDIR, a novel DIR framework based on Implicit Neural Representations (INRs) that conditions the registration process on regularization hyperparameters. Unlike conventional methods that require retraining for each regularization hyperparameter setting, cIDIR is trained over a prior distribution of these hyperparameters, allowing real-time tuning during inference. Additionally, it models a continuous and differentiable DVF, enabling seamless integration of advanced regularization techniques. Evaluated on the DIR-LAB (5; 4) dataset, cIDIR achieves high accuracy and robustness across the dataset by leveraging real-time hyperparameter optimization after training.

## 1 Introduction

Deformable image registration (DIR) is essential in medical imaging for aligning images across different modalities, or patients. The effectiveness of DIR relies on enforcing the diffeomorphism of the applied transformations (15), as it ensures physical plausibility of deformation. To enforce these properties, various regularization strategies have been proposed (13; 3). Recent advances in deep neural networks have led to the introduction of numerous DIR approaches (8; 11; 1). These methods learn to predict DVF on a grid for unseen image pairs. However, these grid-based methods provide a discontinuous representation, making it challenging to incorporate advanced regularization techniques that require accurate gradient computations. Therefore, the need to incorporate these regularization has driven the development of methods that learn continuous representations of the DVF. A class of method is based on implicit neural representations (INRs) (12). Wolterink *et al.* (16) introduced IDIR, an implicit DIR model that seamlessly integrates regularization techniques. While the method demonstrated high accuracy when using the Bending Energy regularization (14), it requires hyperparameter tuning to balance the weights between data and regularization losses. Standard hyperparameter optimization methods, such as random, grid, and sequential search (2), are commonly used for this aim. However, these methods are computationally intensive, as they require retraining the model multiple times to assess each hyperparameter choice. In this work, we build upon IDIR by introducing cIDIR, a simple, yet effective approach that conditions an INR of the DVF on the hyperparameters of the loss functions. Similar to IDIR, cIDIR offers a continuous and differentiable representation of the DVF, facilitating the integration of regularization techniques. The key difference, however, lies in how hyperparameter tuning is handled. While IDIR requires multiple training sessions to optimize the hyperparameters, cIDIR only needs to be trained once. Hyperparameter tuning is performed during inference, allowing for real-time adjustments.

Submitted to 39th Conference on Neural Information Processing Systems (NeurIPS 2025). Do not distribute.

## 2 Methods

We introduce a learning-based framework called cIDIR, designed to learn an implicit representation of the deformation vector field (DVF) in a flexible and adaptive manner. Figure 1 provides an overview of the framework. cIDIR is composed of two main components: the **main network** and the **harmonizer network**. The **main network** is responsible for learning the implicit representation of the DVF, while the **harmonizer network** conditions this representation on the regularization weighting factor $\alpha$. Both networks are trained in an end-to-end fashion, with $\alpha$ uniformly sampled from the range $[0, 1]$ during training to ensure robust conditioning and generalization.

The **main network** in **cIDIR** is implemented as a multilayer perceptron (MLP) with input and output dimensions of 3, corresponding to the spatial coordinates of the moving and fixed images. It comprises three hidden layers, each containing 256 neurons followed by a non-linear activation function. Since the network is designed as an INR, the selection of the activation function is critical for effectively capturing spatial variations. Standard activation functions such as ReLU are known to bias networks toward low-

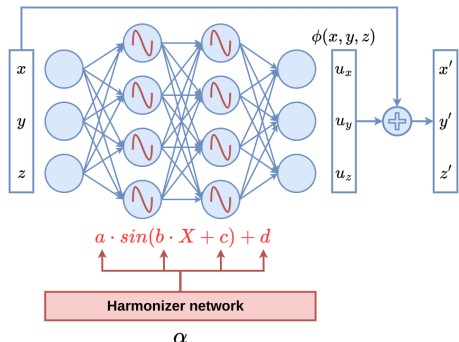

Figure 1: Overview of cIDIR. The main network (in blue) learns an implicit representation of the DVF $\phi$, mapping coordinates $(x, y, z)$ from the moving to $(x', y', z')$ in the fixed image. The network is conditioned on the regularization weighting factor $\alpha$ through the harmonization network, which predicts the parameters $a$, $b$, $c$, and $d$ of the activation function used in the main network.

frequency signals (6), making them unsuitable for modeling detailed deformations. To overcome this limitation, prior works have proposed the use of periodic activation functions (16), which allow the network to represent high-frequency components more effectively. In our approach, we employ a parameterized activation function $\sigma$, inspired by (10), and defined as: $\sigma(x) = a \cdot \sin(b \cdot x + c) + d$. The parameters of this function control distinct aspects of the network's response. Specifically, $a$, $b$, $c$, and $d$ control the shape of the sinusoidal function. By learning these parameters, the activation function becomes flexible and adaptive, enabling to capture complex deformation fields with varying levels of spatial detail. The **harmonizer network** serves as a conditioning mechanism that links the main network's behavior to the regularization weighting factor $\alpha$. This network is designed as an MLP that takes $\alpha$ as input and predicts four outputs corresponding to the activation parameters of the main network. The harmonizer consists of three hidden layers with sizes **128**, **64**, and **32**, respectively. Each hidden layer is followed by **layer normalization** to stabilize training and enhance generalization, as well as a SiLU activation function, which promotes smooth gradient propagation.

Upon **cIDIR**'s training, a grid search over values of $\alpha$ in the range $[0, 1]$ is performed. For each $\alpha$, a displacement field is generated and applied to the moving image to produce a **moved image**. Both the **moved and fixed images** are then segmented using the method from (9), and the resulting segmentations are used to compute a Dice score (DS). The $\alpha$ value that yields the highest DS is selected as the optimal. **The segmentations serve as observations to guide the optimization of $\alpha$.**

## 3 Results

We validate our approach on the DIR-LAB 4DCT dataset, which includes 10 patient scans. For each case, **cIDIR** was trained for 50K epochs using Bending Energy regularization, with the weight $\alpha$ sampled uniformly in $[0, 1]$. During training, 10K points were randomly drawn from the segmented lung region of the moving image (9), and the model was optimized using the Normalized Cross-Correlation (NCC) loss. After training, the optimal $\alpha$ was determined using the segmentation as an observation. As shown in Table 1, **cIDIR** achieves lower average Target Registration Error (TRE) than state-of-the-art methods, including **IDIR** (16) and **CNN** (7). Unlike **IDIR**, which requires retraining for each regularization weight (fixed at $\alpha = 10$), **cIDIR** can select the optimal $\alpha$ in real time after a single training, providing improved accuracy and efficiency across patients. To further evaluate **cIDIR**'s adaptability across different regularization techniques, we conducted experiments using both hyperelasticity and Jacobian regularizations. For **IDIR**, we trained the

model with a fixed value of $\alpha = 0.5$ across all $10$ patients. **cIDIR** was trained on a uniform distribution of $\alpha$ over $[0, 1]$, and at inference, we selected $\alpha = 0.5$. As shown in Table 2, both methods achieved comparable results for the Jacobian regularization, while **cIDIR** outperformed **IDIR** with hyperelasticity.

Table 1: TRE in mm of **cIDIR** compared to learning-based methods: **IDIR** (16), **CNN** (7), and **Displacement** (TRE before registration). Both **IDIR** and **cIDIR** use Bending Energy regularization, with $\alpha = 10$ for **IDIR** as proposed in their paper.

| Scan | cIDIR (ours) | IDIR (16) | CNN (7) | Displacement |
|---|---|---|---|---|
| 4DCT 01 | 0.66 (1.25) | 0.52 (1.11) | 1.21 (0.88) | 4.01 (2.91) |
| 4DCT 02 | 0.76 (1.33) | 0.55 (1.15) | 1.13 (0.65) | 4.65 (4.09) |
| 4DCT 03 | 0.68 (1.23) | 0.76 (1.32) | 1.32 (0.82) | 6.73 (4.21) |
| 4DCT 04 | 1.18 (1.3) | 0.82 (1.47) | 1.84 (1.76) | 9.42 (4.81) |
| 4DCT 05 | 1.17 (1.86) | 1.29 (1.78) | 1.80 (1.60) | 7.10 (5.14) |
| 4DCT 06 | 0.82 (1.84) | 0.86 (1.40) | 2.30 (3.78) | 11.10 (6.98) |
| 4DCT 07 | 1.35 (1.65) | 1.76 (2.29) | 1.91 (1.65) | 11.59 (7.87) |
| 4DCT 08 | 1.44 (3.05) | 2.54 (4.30) | 3.47 (5.00) | 15.16 (9.11) |
| 4DCT 09 | 3.72 (2.59) | 3.54 (2.65) | 1.47 (0.85) | 7.82 (3.99) |
| 4DCT 10 | 1.61 (2.14) | 1.50 (1.94) | 1.79 (2.24) | 7.63 (6.54) |
| **Average** | **1.33** | 1.47 | 1.83 | 8.52 |

Table 2: Comparison of TRE in for **cIDIR** and **IDIR** on the DIR-LAB (5; 4) dataset using Hyperelastic (3) and Jacobian (13) regularizations. **IDIR** is trained with a fixed $\alpha = 0.5$, while **cIDIR** is trained over $\alpha \in [0, 1]$ and evaluated with $\alpha = 0.5$.

| Scan | Hyperelastic ($\alpha = 0.5$) | | Jacobian ($\alpha = 0.5$) | |
|---|---|---|---|---|
| | cIDIR | IDIR | cIDIR | IDIR |
| 4DCT 01 | 0.73 (1.16) | 7.10 (5.48) | 1.46 (1.77) | 1.61 (2.18) |
| 4DCT 02 | 0.63 (1.14) | 3.34 (3.09) | 2.78 (2.31) | 2.55 (3.06) |
| 4DCT 03 | 1.32 (1.80) | 3.99 (2.81) | 4.11 (2.78) | 3.17 (3.59) |
| 4DCT 04 | 1.11 (1.44) | 7.83 (4.90) | 5.65 (4.65) | 3.95 (5.72) |
| 4DCT 05 | 1.42 (1.79) | 6.35 (5.36) | 2.86 (3.13) | 2.29 (2.99) |
| 4DCT 06 | 1.08 (1.35) | 8.70 (6.64) | 3.68 (3.32) | 5.95 (6.99) |
| 4DCT 07 | 2.48 (2.48) | 6.97 (7.37) | 6.40 (5.32) | 6.20 (5.18) |
| 4DCT 08 | 6.00 (5.71) | 7.89 (7.98) | 11.4 (10.9) | 11.01 (11.51) |
| 4DCT 09 | 3.59 (3.10) | 7.23 (6.02) | 3.97 (2.41) | 3.15 (2.49) |
| 4DCT 10 | 2.17 (2.09) | 5.17 (4.52) | 6.91 (4.4) | 6.26 (5.34) |
| **Average** | 2.05 | 6.45 | 4.92 | 4.61 |

# 4 Conclusion

In this work, we introduced **cIDIR**, it leverages an INR to model a continuous deformation vector field, enabling the integration of advanced regularization techniques that require higher-order gradients. By conditioning the activation functions of the INR on the regularization weighting factor, **cIDIR** allows for real-time hyperparameter optimization after training, eliminating the need for expensive retraining. Our experiments highlight **cIDIR**'s accuracy, computational efficiency, and robustness across different regularization techniques. However, **cIDIR** has limitations. Its patient-specific nature requires a dedicated training phase for each new subject, leading to long training times that may hinder its practical deployment, particularly in time-sensitive clinical settings. Another limitation is the assumption of a well-defined prior distribution for regularization parameters, which may not always align with the optimal settings for every case. Future work will explore strategies to improve generalization across patients and datasets, and extending **cIDIR** to broader applications, such as multi-modal image registration. Finally, expanding **cIDIR** to handle multiple hyperparameters could allow for the efficient integration of diverse regularization techniques within a single training process, further strengthening diffeomorphism enforcement.

**Aknowledgments** This work was financially supported by the Werner Siemens Foundation through the MIRACLE II project.

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
