# OpenReview forum: "Conditioned Implicit Neural Representation for Regularized Deformable Image Registration"
_EurIPS.cc/2025/Workshop/MedEurIPS — EurIPS 2025 Workshop MedEurIPS Submission_

### Official Review · Reviewer_bt6G · 2025-10-28
**cIDIR review comments**

**Rating:** 6
**Confidence:** 5

**Review:**

This paper presents a novel method based on implicit neural representation condition for deformable image registration. The main network predicts the moved image coordinates and the additional harmonizer network predicts the activation parameters of main network. This forms a regularization that operates during inference stage for real-time adjustments to guarantee a smooth and physical plausible deformation.

Major concerns:
1. The evaluation is limited by using a small, private dataset. I strongly encourage the authors to include additional results on large-scale public medical image registration datasets to validate generalizability.
2. The paper lacks sufficient qualitative analysis. Include visualizations of the registration quality, such as pixel-level registration error maps, to better demonstrate performance.
3. The method's generalizability should be better demonstrated. Consider showing applications to a broader range of tasks, including registration of other organs image and multi-modality image alignment.

---

### Official Review · Reviewer_3THQ · 2025-10-31
**Well structured and novel paper, with a potential for improvement in motivation**

**Rating:** 7
**Confidence:** 4

**Review:**

This paper presents a concise novel extension of an existing method. They are the first to combine two methods: the already established INR approach for registration and conditioning on regularization terms.

This contribution would meaningfully contribute to the workshop, e.g. as a starting point of how to integrate methods from various fields into an imporved new approach.

The authors clearly state the problem and then solve it.

The methodology and results are clear and the figures support the understanding of their methods.


What could be improved:

- the motivation of conditioning on regularization terms seems to be different in INRs from CNNs - it should be better explained for INR. To my understanding, CNNs are trained on an entire dataset and then on the test set it makes sense to want to adjust the regularization terms on the fly. However, with INRs, a new network is trained for each new image pair, so the desired characteristics of the DVF for this image pair can be controlled directly during training. Also, even though it is not mentioned in the paper, training an INR with a harmonizer network probably also increase the training time - and finally in their approach, multiple inference runs are peformed with varying regularization terms, which again adds to the compute time.

---

### Decision · Program_Chairs · 2025-10-31

**Decision:**

Accept (Poster)

**Comment:**

Both reviewers find the paper novel and well structured, highlighting its clear methodology and contribution.